# Impact of MAOA Gene Polymorphism on the Efficacy of Antidepressant Treatment and Craving Severity for Betel Quid Use Disorder

**DOI:** 10.3390/ijms25179221

**Published:** 2024-08-25

**Authors:** Chung-Chieh Hung, Ying-Chin Ko, Ping-Ho Chen, Chia-Min Chung

**Affiliations:** 1School of Medicine, Chung Shan Medical University, Taichung 40201, Taiwan; cshy2135@csh.org.tw; 2Department of Psychiatry, Chung Shan Medical University Hospital, Taichung 40201, Taiwan; 3Department of Medical Research, China Medical University Hospital, Taichung 40447, Taiwan; ycko0406@gmail.com; 4Graduate Institute of Toxicology, College of Medicine, National Taiwan University, Taipei 10051, Taiwan; 5School of Dentistry, College of Dental Medicine, Kaohsiung Medical University, Kaohsiung 80708, Taiwan; phchen@kmu.edu.tw; 6Institute of Biomedical Sciences, National Sun Yat-Sen University, Kaohsiung 80424, Taiwan; 7Department of Veterinary Medicine, College of Veterinary Medicine, National Pingtung University of Science and Technology, Pingtung 91201, Taiwan; 8Department of Medical Research, Kaohsiung Medical University Hospital, Kaohsiung 80756, Taiwan; 9Center for Medical Informatics and Statistics, Office of Research and Development, Kaohsiung Medical University, Kaohsiung 80708, Taiwan; 10Graduate Institute of Biomedical Sciences, China Medical University, Taichung 40402, Taiwan; 11Department of Psychiatry and Center for Addiction and Mental Health, China Medical University Hospital, Taichung 40447, Taiwan

**Keywords:** betel quid use disorder, craving severity, MAOA gene polymorphism, personalized medicine, randomized clinical trial

## Abstract

Betel quid (BQ) use disorder (BUD) is prevalent in many Asian countries, impacting approximately 600 million people. We conducted a randomized clinical trial to analyze the impact of MAOA genetic variations on the severity of BQ craving. This was measured using *DSM-5* criteria and the Yale–Brown Obsessive–Compulsive Scale modified for betel quid use (Y-BOCS-BQ). Participants were grouped according to the severity of BUD and MAOA gene single-nucleotide polymorphism (SNP) rs5953210 genotypes. The Y-BOCS-BQ scores were assessed at baseline (week 0) and during follow-up at weeks 2, 4, 6, and 8. The AA genotype group showed significantly greater reductions in Y-BOCS-BQ at weeks 2 (*p* = 0.0194), 4 (*p* = 0.0078), 6 (*p* = 0.0277), and 8 (*p* = 0.0376) compared to the GG genotype group. Additionally, within the antidepressant group, the AA genotype showed significant reductions in the Y-BOCS-BQ scores at weeks 2 (*p* = 0.0313), 4 (*p* = 0.0134), 6 (*p* = 0.0061), and 8 (*p* = 0.0241) compared to the GG genotype. The statistical analysis revealed a significant interaction between the treatment and placebo groups based on MAOA genotypes, with the AA genotype in the treatment group exhibiting a more pronounced decrease in Y-BOCS-BQ score (*p* interaction <0.05) at week 6. Our study highlights the importance of considering genetic factors when developing personalized treatment plans for BUD.

## 1. Introduction

Betel quid (BQ) use disorder (BUD) is a prevalent addiction disease in many Asian countries [1], and its presentations can be defined and derived from the *Diagnostic and Statistical Manual of Mental Disorders, Fifth Edition (DSM-5)* [2], affecting approximately 600 million people globally [3]. The symptoms and signs of BUD according to *DSM-5* can be classified into four categories, including impaired control, social impairment, risky use, and pharmacological indicators concerning the people who become addicted to betel quid (BQ) consumption [1,4]. BQ, the fourth most popular psychoactive substance [5], is a group I human carcinogen that significantly increases the risk of oral potentially malignant disorders (OPMDs) [6] and cancers of the oral cavity and pharynx [7,8]. This carcinogenic risk related to BQ consumption is particularly higher in regions like Taiwan, where oral and pharyngeal cancers are among the most common etiologies of morbidity and mortality in men [9].

The addictive nature and property of BQ is not entirely understood, but it is known to interact with monoamine oxidase inhibitors (MAOIs), impacting cellular neurotransmitter levels in the brain [10]. One of the key components of BQ, arecoline, exhibits MAOA-inhibitor-like properties, preventing the breakdown of neurotransmitters and increasing the concentrations of dopamine and serotonin in the brain. This mechanism is similar to how certain antidepressants, such as MAOIs and SSRIs, function. Antidepressant drugs, including MAOIs and SSRIs, can reduce AN use in mice [11], potentially influencing BQ consumption among individuals with depression. Studies have shown that antidepressant therapy with MAOIs and SSRIs can decrease daily BQ use in patients with depression [12,13].

Genetic MAOA variations are associated with heavy BQ use, which positively correlate with MAOA enzymatic activity levels in the human plasma [14]. Cell and animal models have revealed that areca nut (AN) and arecoline inhibit MAOA mRNA and protein expression and have MAOA inhibitor-like properties. However, multiple gene variants in combination due to the genetic complexity of addiction diseases can explain a larger proportion of the genetic contribution to BUD, which will allow for improved treatment strategies of addiction diseases [15].

Previously, clinical evidence supported using antidepressants to treat BUD. Retrospective studies suggested that antidepressants can reduce BQ consumption and addiction severity in depressed patients. A clinical trial showed that antidepressants could effectively decrease BQ use within 2–4 weeks [13]. However, individual responses to treatment can vary significantly due to genetic differences, affecting the metabolism and efficacy of antidepressants in reducing BQ use, rendering different outcomes for patients with BUD.

Currently, there are no efficient drugs for the treatment of BUD. No reliable biomarkers exist for the diagnosis of BUD or serve as predictive values of its treatment. Hence, we conducted the first clinical trial to investigate the response to antidepressant treatment for BUD in relation to MAOA gene polymorphism. The primary objective of this study was to enhance BUD therapy by developing reliable biomarkers. Specifically, we examined the impact of MAOA genetic variations on the severity of betel quid craving. Incorporating genetic biomarkers can improve diagnostic accuracy and enable personalized treatment by considering genetic differences among patients.

## 2. Results and Discussion

### 2.1. Basic Demographic Information of the Study Subjects

In Table 1, the mean age, frequency of BQ, and amount of BQ consumption in the treatment group were 44.34 years (8.93), 5.54 days per week (2.33), and 50.97 (55.73), respectively, while in the placebo group, they were 43.07 (8.40), 6.13 (1.81), and 36.27 (39.64), which were not statistically significantly different. Regarding the *DSM-5* criteria for BQ, tobacco, and alcohol, as well as SUSRS scores, there were no significant differences between the two groups. Additionally, the Y-BOCS-BQ scores showed no significant differences between the two groups from the baseline to week 8 of the clinical trial.

### 2.2. Association between Severity of BUD and Follow-Up Y-BOCS-BQ Score

At week 0 in Table 2, the severe BUD group had a mean Y-BOCS-BQ score of 31.6 (8.5), compared to 22.7 (8.3) in the non-severe BUD group, with a *p*-value of 0.0011. At week 2, the severe BUD group scored 24.6 (14.7) versus 16 (11.6) in the non-severe BUD group, with a *p*-value of 0.0457. At week 4, the severe BUD group scored 20.5 (16.6), while the non-severe BUD group scored 11.2 (10.1), with a *p*-value of 0.0272. Significant differences in Y-BOCS-BQ scores were observed between the severe and non-severe BUD groups at weeks 0, 2, and 4 but not at weeks 6 and 8.

### 2.3. Association between MAOA Gene Polymorphism (rs5953210) and Follow-up Y-BOCS-BQ Scores in Patients with Betel Quid Use Disorder 

The data in Table 3 compare the mean Y-BOCS-BQ scores between the two genotype groups: AA (*n* = 22) and GG (*n* = 28). No significant difference between the two genotype groups was observed at week 0. However, the AA genotype was associated with significantly greater reductions in Y-BOCS-BQ scores at weeks 2 (*p* = 0.0194), 4 (*p* = 0.0078), 6 (*p* = 0.0277), and 8 (*p* = 0.0376) compared to the GG genotype. Specifically, the craving score in the AA genotype group decreased from 27.0 at week 0 to 11.0 at week 8. In contrast, the craving score in the GG genotype group decreased from 30.2 at week 0 to 21.2 at week 8.

### 2.4. Differential Treatment Response Based on Y-BOCS-BQ Scores and Severity of BUD in Antidepressant and Placebo Groups

In the antidepressant group in Table 4, there was a significant difference in Y-BOCS-BQ scores at the baseline (week 0) between the severe and non-severe groups (*p* = 0.0006). However, after the initiation of antidepressant treatment, Y-BOCS-BQ scores did not show significant differences during the follow-up period. In the placebo group, no significant differences were observed in Y-BOCS-BQ scores between the severe and non-severe groups at any follow-up week.

### 2.5. Differential Effect of MAOA Gene Polymorphism on Antidepressant Treatment Efficacy and Craving Severity in BUD

In Table 5, the AA genotype in the treatment group shows significant reductions in Y-BOCS-BQ scores at weeks 2 (*p* = 0.0313), 4 (*p* = 0.0134), 6 (*p* = 0.0061), and 8 (*p* = 0.0241) compared to the GG genotype. However, no significant differences were observed in the placebo group at any time point.

Additionally, Appendix A illustrate the interaction between MAOA genotypes and follow-up Y-BOCS-BQ scores. To compare the treatment and placebo groups at week 6 (Appendix A), the data points are plotted for two genotypes: AA (blue) and GG (red). And statistical analysis revealed a significant interaction between the treatment and placebo groups based on MAOA genotype. The treatment group exhibited a more pronounced decrease in Y-BOCS-BQ scores for the AA genotype compared to the GG genotype (*p* interaction < 0.05). In contrast, the placebo group did not show significant changes.

### 2.6. Discussion

Our study investigated the effects of the MAOA gene SNP rs5953210 on BUD treatment outcomes using Y-BOCS-BQ scores. The results reveal significant gene and treatment interactions. Subjects with the AA genotype of rs5953210 exhibited greater reductions in Y-BOCS-BQ scores over time compared to those with the GG genotype, particularly in the treatment group. No significant differences were observed between the AA and GG genotypes in the placebo group. The statistical analysis confirmed significant differences in Y-BOCS-BQ scores between the severe and non-severe BUD groups at weeks 0, 2, and 4 but not at weeks 6 and 8. The treatment group, especially those with the AA genotype, showed significant reductions in Y-BOCS-BQ scores at weeks 2, 4, 6, and 8 compared to the GG genotype, indicating a more pronounced decrease in craving symptoms.

Polymorphisms of the MAOA gene are associated with different types of addictive substances and illicit drugs, with varying results across different populations and substances. The MAOA variable number tandem repeat (VNTR) polymorphism has been studied in relation to alcohol use disorder [16,17,18,19], as well as various types of substance and drug addictions [20,21,22]. Fite et al. (2019) found that MAOA VNTR variants influence polysubstance use, with this relationship being moderated by emotional or physical abuse during childhood in a sex-specific manner [23]. Sun et al. (2017) identified an association between the MAOA rs1137070 C allele and heroin addiction in Chinese individuals [24]. Conversely, Chien et al. (2010) found no link between the MAOA promoter VNTR polymorphism and heroin addiction in Chinese men [22]. Hung et al. (2024) identified an association between the MAOA rs5953210 variant and patients with severe BUD [15]. The underlying mechanisms through which the MAOA gene is associated with substance use disorders and illicit drug use are not yet fully understood but are likely due to overlapping neurobiological pathways.

Monoamine oxidases (MAOs) A and B are mitochondrial-bound isoenzymes that catalyze the oxidative deamination of dietary amines and neurotransmitters such as serotonin, norepinephrine, dopamine, beta-phenylethylamine, and other trace amines [25]. This rapid degradation is crucial for maintaining proper synaptic neurotransmission and regulating emotional behaviors and brain functions [26]. MAOA, primarily found in dopaminergic neurons, and MAOB, mainly expressed in serotonergic neurons, both contribute to the etiology of addiction disorders [27]. The dopamine system plays a central role in the biology of BUD. Cell and animal models have shown that AN and arecoline inhibit MAOA mRNA and protein expression, exhibiting monoamine oxidase inhibitor (MAOI)-like properties [14]. Arecoline primarily increases serotonin levels, likely through MAOA inhibition, preventing neurotransmitter breakdown and thereby increasing dopamine and serotonin concentrations in the brain [28]. AN produces potential antidepressant effects by elevating serotonin and noradrenaline levels. Thus, the use of MAOIs may have clinical benefits for BQ cessation among heavy BQ users. Antidepressant therapy has been observed to reduce daily BQ use in patients with depression [13]. Currently, no pharmacologically based cessation therapies are available to alleviate symptoms in patients with BUD who intend to reduce or quit BQ use. Our study found that different MAOA genotypes exhibit varying responses to antidepressant treatment. This finding enables the identification of individuals who respond well to antidepressant therapy. Since not all patients seem to benefit from antidepressants, the use of genetic testing might be important for improving the effectiveness of treatment with MAOA inhibitors.

### 2.7. Study Limitations

There are several limitations in this study that warrant consideration. Firstly, our study included only limited phenotypic and genotypic information, indicating the necessity for genotyping additional novel susceptibility genes identified by genome-wide association studies. Secondly, this study is the first randomized clinical trial to evaluate the efficacy of MAOA and SSRI antidepressants for BQ cessation treatment. However, as a pioneering study with a relatively small sample size as a study limitation, it still provides only preliminary evidence, necessitating replication in larger trials to validate these findings.

## 3. Materials and Methods

### 3.1. Study Participants

This study recruited participants from the cancer centers of the Department of Dentistry and the Department of General Physicians at China Medical University Hospital in Taichung, Taiwan, between January 2016 and April 2019. A total of 50 patients with betel quid (BQ) chewing habits were enrolled. Data on their basic demographic characteristics were collected, and the clinical features related to their BQ addiction were assessed. All participants provided informed consent and underwent clinical interviews conducted by a psychiatrist. BQ use disorder (BUD) was diagnosed based on the *Diagnostic and Statistical Manual of Mental Disorders, Fifth Edition* (*DSM-5*) criteria. The diagnosis of BUD was determined by the presence of at least 2 of the following 11 symptoms within the past one year: (1) extensive or prolonged BQ consumption, (2) unsuccessful attempts to reduce BQ use, (3) significant time spent chewing, (4) cravings, (5) neglect of major responsibilities, (6) social or interpersonal issues, (7) abandoning activities, (8) hazardous use, (9) continued use despite awareness of problems, (10) tolerance, and (11) withdrawal. BUD severity was classified as mild (2–3 symptoms), moderate (4–5 symptoms), or severe (≥6 symptoms). In our study design, subjects with severe BUD were considered as having more than 6 of the *DSM-5* symptoms for BUD while subjects non-severe BUD possessed less than 6.

Participants were excluded if they (1) abused illegal substances (e.g., heroin, amphetamines), (2) had major psychiatric disorders (e.g., schizophrenia, bipolar disorder, major depressive disorder, antisocial personality disorder), (3) had organic brain conditions (e.g., cerebrovascular disease, brain tumor, head injury), (4) had any form of cancer or cancer-related disease, or (5) were unable to understand or speak Chinese. The study psychiatrist conducted semi-structured diagnostic interviews and systemic reviews for psychiatric and addictive disorders [29,30,31]. Cancer diagnoses were confirmed and excluded in this study at the study hospital’s cancer center (e.g., cancers in the nasopharyngeal organs, gastrointestinal organs, and lungs), while neurological or other brain disorders were identified based on patient self-reports of their medical histories. Patients diagnosed with anxiety disorders or sleep disturbances and those with habits of consuming alcohol, cigarettes, caffeine, or hypnotics were not excluded if their hypnotic dosage had been consistent over the past year and they did not meet more than 6 criteria in the *DSM-5* for other substance use disorders in the past one year.

Information regarding the initial age of BQ consumption, daily BQ consumption amount, and weekly consumption frequency was collected for all patients with BUD. Oral hygiene was assessed using the visual analog scale (VAS), and data on the number of broken teeth and daily tooth brushing frequency were obtained through self-reports. This study was approved by the China Medical University and Hospital, Chung Shan Medical University Hospital Research Ethics Committee (CMUH103-REC1-059, CMUH106-REC1-016, CSMUH No: CS1-23163).

### 3.2. Psychometric Measures of Addiction Severity

To establish a definitive diagnosis and assess the severity of BQ use disorder (BUD), we applied and employed previously validated *DSM-5* criteria [1]. The Substance Use Severity Rating Scale (SUSRS) was utilized to evaluate BQ and alcohol consumption, as well as cigarette-smoking habits. The SUSRS, developed based on the *DSM-IV* and the *International Classification of Diseases, Eleventh Revision* (*ICD-11*) [30], comprises 21 items that measure the severity of substance use addiction. It has been widely applied in assessing alcohol consumption, cigarette smoking, and drug use [32,33]. Smoking is one of the confounding factors in our study since smoking can impact the symptom severity of psychiatric problems [34]. In this study, a rater assessed participants’ substance use with yes-or-no questions, assigning a score of 1 for “yes” and 0 for “no”. Additionally, the Yale–Brown Obsessive–Compulsive Disorder Rating Scale for betel quid (Y-BOCS-BQ) was used to determine the severity of BQ craving [35,36]. The Y-BOCS-BQ was specifically designed to measure the behavioral problems associated with BQ use [37,38] and is commonly employed to analyze the severity of cravings in substance abuse [39].

### 3.3. DNA Extraction and Genotyping

The participants’ genomic DNA was extracted from peripheral blood samples using a Puregene DNA Isolation Kit (Gentra Systems, Minneapolis, MN, USA) following the manufacturer’s instructions. Genotyping of the MAOA SNPs was performed using the Sequenom MassARRAY System at the Academia Sinica National Genotyping Center (Taipei, Taiwan).

### 3.4. Methods of Statistical Analysis

The statistical analysis was performed using SAS 9.4 software (Cary, NC, USA). Genotype frequencies in the control population were tested for Hardy–Weinberg equilibrium, with differences between observed and expected genotype numbers compared using the chi-square test. Hardy–Weinberg equilibrium was assumed for *p*-values > 0.05. The *t*-tests were applied to compare the demographic information of patients with BUD and the conditions of antidepressant treatment. Since no healthy controls were included in this study, we combined the mild and moderate BUD groups into a non-severe BUD group, which served as the statistical control. A logistic regression analysis model was used to investigate the association between SNPs and the severity of BUD. Mixed models and General Linear Models were used to compare the means and interactions of quantitative variables between the genotypes.

## 4. Conclusions

These findings highlight the critical role of genetic factors in developing medical therapies for BUD, emphasizing the interaction between MAOA genotypes and antidepressant treatment in mitigating craving scores. Customizing treatment strategies based on individual genetic profiles can significantly enhance the efficacy of interventions for BUD. Identifying genetic biomarkers would improve the accuracy of BUD diagnosis and facilitate personalized treatment approaches by accounting for genetic differences among patients.

## Figures and Tables

**Table 1 ijms-25-09221-t001:** Basic characteristics of study subjects and follow-up of Y-BOCS-BQ score.

Variables	Treatment Group (*n* = 35)	Placebo Group (*n* = 15)	*p*-value
	Mean (SD)	Mean (SD)	
Age	44.34 (8.93)	43.07 (8.40)	0.64
Days of BQ Consumption	5.54 (2.33)	6.13 (1.81)	0.39
BQ Amount	50.97 (55.73)	36.27 (39.64)	0.36
*DSM-5* BQ	6.57 (2.50)	6.80 (2.54)	0.77
*DSM-5* Tobacco	7.26 (2.80)	6.00 (3.85)	0.20
*DSM-5* Alcohol	2.74 (3.62)	2.47 (3.58)	0.81
SUSRS BQ	13.31 (5.40)	13.47 (4.02)	0.92
SUSRS Tobacco	13.60 (4.53)	12.67 (5.65)	0.54
SUSRS Alcohol	6.34 (6.83)	6.00 (7.76)	0.88
Y-BOCS-BQ Week 0	28.37 (9.84)	29.73 (8.48)	0.64
Y-BOCS-BQ Week 2	21.74 (15.61)	22.07 (10.91)	0.94
Y-BOCS-BQ Week 4	18.20 (15.70)	15.93 (9.38)	0.61
Y-BOCS-BQ Week 6	18.00 (16.10)	15.87 (15.04)	0.66
Y-BOCS-BQ Week 8	15.97 (17.79)	18.40 (16.60)	0.65

Abbreviation: BQ: betel quid. *DSM-5*: *Diagnostic and Statistical Manual of Mental Disorders, Fifth Edition*. SUSRS: Substance Use Severity Rating Scale. Y-BOCS-BQ: Yale–Brown Obsessive–Compulsive Scale modified for betel quid. Scores are presented as means (SD: Standard Deviation). *p*-values indicate the significance of the difference between the treatment and placebo groups for each variable.

**Table 2 ijms-25-09221-t002:** Association between the severity of BUD and Follow-Up Y-BOCS-BQ score.

Variables	Severe BUD (*n* = 34)	Non-Severe BUD (*n* = 16)	*p*-value
Y-BOCS-BQ Week 0	31.6 (8.5)	22.7 (8.3)	0.0011 **
Y-BOCS-BQ Week 2	24.6 (14.7)	16 (11.6)	0.0457 *
Y-BOCS-BQ Week 4	20.5 (16.6)	11.2 (10.1)	0.0272 *
Y-BOCS-BQ Week 6	17.7 (16.6)	16.6 (14)	0.8227
Y-BOCS-BQ Week 8	18.7 (18.4)	12.4 (14.4)	0.2292

Abbreviations: BQ: betel quid. BUD: betel quid use disorder. Y-BOCS-BQ: Yale–Brown Obsessive–Compulsive Scale modified for betel quid. Scores are presented as means (SD: Standard Deviation). *p*-values indicate the significance of the difference in scores between severe and non-severe BUD groups at each time point of visits. * *p* < 0.05 and ** *p* < 0.01, statistical significance between groups.

**Table 3 ijms-25-09221-t003:** Association between MAOA gene polymorphism (rs5953210) and follow-up Y-BOCS-BQ scores in patients with betel quid use disorder.

	rs5953210		
Variables	AA (*n* = 22)	GG (*n* = 28)	
	Mean (SD)	Mean (SD)	*p*-value
Y-BOCS-BQ Week 0	27.0 (9.8)	30.2 (8.9)	0.2266
Y-BOCS-BQ Week 2	16.6 (13.1)	26.0 (13.9)	0.0194 *
Y-BOCS-BQ Week 4	11.7 (12.1)	22.1 (13.9)	0.0078 **
Y-BOCS-BQ Week 6	11.9 (13.7)	21.6 (15.9)	0.0277 *
Y-BOCS-BQ Week 8	11.0 (15)	21.2 (17.7)	0.0376 *

Abbreviations: BQ: betel quid. MAOA: monoamine oxidase A. SNP: single-nucleotide polymorphism. Y-BOCS-BQ: Yale–Brown Obsessive–Compulsive Scale modified for betel quid. Scores are presented as means (SD: Standard Deviation). *p*-values indicate the significance of the difference in scores between AA and GG genotypes at each time point of visits. * *p* < 0.05 and ** *p* < 0.01, statistical significance between groups.

**Table 4 ijms-25-09221-t004:** Differential treatment response based on Y-BOCS-BQ scores and severity of BUD in antidepressant and placebo groups: a randomized clinical trial.

	Treatment Group			Placebo Group		
	Severe (*n* = 23)	Non-Severe (*n* = 12)		Severe (*n* = 11)	Non-Severe (*n* = 4)	
	Mean (SD)	Mean (SD)	*p*-value	Mean (SD)	Mean (SD)	*p*-value
Y-BOCS-BQ Week 0	32.2 (9.3)	21 (6.1)	0.0006 **	30.5 (7.0)	27.8 (12.9)	0.6037
Y-BOCS-BQ Week 2	24.7 (16.5)	16.1 (12.5)	0.1229	24.4 (10.7)	15.8 (9.9)	0.1857
Y-BOCS-BQ Week 4	21.5 (16.8)	11.8 (11.4)	0.0829	18.4 (9.4)	9.3 (5.7)	0.0969
Y-BOCS-BQ Week 6	20.1 (17.9)	14.0 (11.5)	0.2952	12.7 (12.9)	24.5 (19.2)	0.1898
Y-BOCS-BQ Week 8	18.4 (19.4)	11.3 (13.8)	0.2629	19.4 (17.0)	15.8 (17.6)	0.7238

Abbreviations: BQ: betel quid. BUD: betel quid use disorder. Y-BOCS-BQ: Yale–Brown Obsessive–Compulsive Scale modified for betel quid. Scores are presented as means (SD: Standard Deviation). *p*-values indicate the significance of the difference between the severe and non-severe BUD groups within each treatment group (antidepressant and placebo) at each time point. ** *p* < 0.01, statistical significance between groups.

**Table 5 ijms-25-09221-t005:** Differential response of MAOA gene polymorphism on antidepressant treatment efficacy and craving severity in BUD: a randomized clinical trial.

	Treatment Group			Placebo Group		
	AA (*n* = 15)	GG (*n* = 20)		AA (*n* = 7)	GG (*n* = 8)	
	Mean (SD)	Mean (SD)	*p*-value	Mean (SD)	Mean (SD)	*p*-value
Y-BOCS-BQ Week 0	26.2 (10.7)	30 (9.1)	0.2644	28.6 (8.0)	30.8 (9.3)	0.6374
Y-BOCS-BQ Week 2	15.3 (14.7)	26.6 (14.7)	0.0313 *	19.4 (8.9)	24.4 (12.5)	0.4011
Y-BOCS-BQ Week 4	10.8 (13.7)	23.8 (15.0)	0.0134 *	13.6 (8.0)	18 (10.5)	0.3814
Y-BOCS-BQ Week 6	9.7 (11.4)	24.3 (16.5)	0.0061 **	16.7 (17.7)	15.1 (13.5)	0.8469
Y-BOCS-BQ Week 8	8.3 (13.5)	21.8 (18.7)	0.0241 *	16.9 (18.3)	19.8 (16.0)	0.7498

Abbreviations: BQ: betel quid. BUD: betel quid use disorder. MAOA: monoamine oxidase A. Y-BOCS-BQ: Yale–Brown Obsessive–Compulsive Scale modified for betel quid. Scores are presented as means (SD: Standard Deviation). *p*-values indicate the significance of the difference in scores between AA and GG genotypes within each treatment group (antidepressant and placebo) at each time point of visits. * *p* < 0.05 and ** *p* < 0.01, statistical significance between groups.

## Data Availability

Data is unavailable due to privacy or ethical restrictions.

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
