# Peer review of "Impact of MAOA Gene Polymorphism on the Efficacy of Antidepressant Treatment and Craving Severity for Betel Quid Use Disorder"

_ijms, 2024, doi:10.3390/ijms25179221_

Round 1

Reviewer 1 Report

Comments and Suggestions for Authors

Scientifically sound paper, clinically relevant and easy to read/understand. I have minor comments for the authors. Please address in your rebuttal letter and track changes in your manuscript. 

1. please include a description of medical status, psychiatric history, cognitive functioning, and functional functioning (see ABAS for instance) in your GG and AA populations

2. please provide a thorough description of how you checked for non-normality in the distribution of your variables, outliers and missing data management, multicollinearity, covariates. 

3. YBOCS please provide some reference to psychometric validity in similar populations (substance use with mood disorders etc. )

Author Response

Comments 1: [please include a description of medical status, psychiatric history, cognitive functioning, and functional functioning (see ABAS for instance) in your GG and AA populations]

Response 1: Thank you very much for your advice. We added the description of medical status, psychiatric history, cognitive functioning, and functional functioning by DSM5 criteria as well as substance severity rating scale for BQ, tobacco, and alcohol. In addition, the baseline functions related to craving status between SNP rs5953210 AA and GG group were rated by Yale-Brown Obsessive Compulsive Scale modified for betel-quid. All variables between AA and GG were calculated without statistical significance. The detailed comparisons were listed as the table below:

SNP rs5953210

Variables

AA (N=22)

GG (N=28)

p-value

Mean (SD)

Mean (SD)

Age

42.6 (8.9)

45.1 (8.6)

0.3137

Days of Consumption

5.1 (2.5)

6.2 (1.8)

0.0945

BQ Amount daily

38.7 (50.1)

52.7 (52.6)

0.3455

DSM_5_BQ

5.9 (2.3)

7.2 (2.5)

0.065

DSM_5_ Tobacco

6.6 (2.8)

7.1 (3.5)

0.5725

DSM_5_Alcohol

2.5 (3.4)

2.8 (3.7)

0.7224

SUSRS_BQ

11.8 (4.4)

14.6 (5.2)

0.0515

SUSRS_ Tobacco

13.3 (3.9)

13.4 (5.6)

0.952

SUSRS_Alcohol

4.6 (5.8)

7.5 (7.8)

0.1556

Y-BOCS-BQ

27.0 (9.8)

30.2 (8.9)

0.2266

Abbreviations: BQ: betel-quid. SNP: Single Nucleotide Polymorphisms. SUSRS: Substance Use Severity Rating Scale. Y-BOCS-BQ: Yale-Brown Obsessive Compulsive Scale modified for betel-quid; Scores are presented as Mean (SD: Standard Deviation).

Comments 2: [please provide a thorough description of how you checked for non-normality in the distribution of your variables, outliers and missing data management, multicollinearity, covariates. ]

Response 2: Thank you for pointing out this. We used the SAS program to assess non-normality in the distribution of our variables. In SAS, there are four test statistics available for detecting non-normality: the Shapiro-Wilk test (Shapiro & Wilk, 1965), the Kolmogorov-Smirnov test, the Cramer-von Mises test, and the Anderson-Darling test. Additionally, we generated a normal probability plot and a Q-Q plot, which compares the ordered values of our variables with the quantiles of a specific theoretical distribution. Since we did not use multiple linear regression analysis in our study, issues related to multicollinearity and covariates were not applicable.

Comments 3: [YBOCS please provide some reference to psychometric validity in similar populations (substance use with mood disorders etc. )]

Response 3: Thank you very much for your advice. We modified the psychometric validity of YBOCS in Line 277-281. And we added the references related to it in reference 20, 22-26. YBOCS is widely translated and used in Chinese with good reliability and validity to measure the severity of craving in addictive diseases such as alcohol, tobacco  and heroin dependence. It possesses good quality for anxiety disorder such as obsessive compulsive disorder in Chinese population is also noted. In previous clinical trial published by Hung et al. (2020) (reference 13 in this article), YBOCS was used as one of the outcome measures of the efficiency in treatment of betel-quid use disorder. In this study, YBOCS were consistent in the severity of BUD with other psychometric parameters such as criteria from DSM-5 or other substance use severity rating scales. 

Reviewer 2 Report

Comments and Suggestions for Authors

1-)if possible you can summarize Betel quid (BQ) use disorder (BUD) in the abstract as a background information.

2-)you can also define BUD in the beginning of the introduction section.

3-)you can highlight the novelty of study in the abstract if the word count allows.

4-)you can make the introduction section related to your research study .

5-)you can remove unnecessary sentences in the results section as you have already mentioned results in the table.

Table 1 presents the characteristics of study subjects across the Treatment (N=35) and 81
Placebo (N=15) groups. The mean age of the treatment group is 44.34 years (SD=8.93), 82
while the placebo group has a mean age of 43.07 years (SD=8.40), with a p-value of 0.64. 83
The average frequency of BQ use is 5.54 days per week (SD=2.33) in the treatment group 84
and 6.13 days per week (SD=1.81) in the placebo group, with a p-value of 0.39. The mean 85
amount of BQ used in the treatment group is 50.97 (SD=55.73), compared to 36.27 86
(SD=39.64) in the placebo group, with a p-value of 0.36. Regarding DSM 5 criteria for BQ, 87
tobacco, and alcohol, as well as SUSRS

6-)after following you can mention it:

. It has been widely applied in assessing alcohol consumption, cigarette smoking, and drug use[17, 18]. Smoking is one of the confounding factors in our study since smoking can impact symptom severity of psychiatric problems (Uludag&Zhao, 2023).

Uludag, K., & Zhao, M. (2023). A narrative review on the association between smoking and schizophrenia symptoms. J Clin Basic Psychosom, 1(1), 1014. 7-)if possible you can write more about inclusion and exclusion criteria of your study. 8-)you can mention low sample size as one of the limitations of your study. 9-)you can add figures related to your study. 10-)you should make explanations about following as you claimed they are not cancer patients. Also, if possible explain how many cancer patients were excluded: The study recruited participants from the cancer centers of the Department of Dentistry and the Department of General Physicians at China Medical University Hospital inTaichung, Taiwan, between January 2016 and April 2019.

Author Response

Comments 1: [if possible you can summarize Betel quid (BQ) use disorder (BUD) in the abstract as a background information.]
Response 1: Thank you very much for your advice. It's really a good suggestion, however, owing to the limitation of 200 words  in the abstract, we'd like to introduce BUD and summarized in the introduction part.

Comments 2: [you can also define BUD in the beginning of the introduction section.]
Response 2: Thank you again for your appreciated advice and we've modified it in Line 45-51.

Comments 3: [you can highlight the novelty of study in the abstract if the word count allows.]
Response 3: Thanks for the advice and we've modified it in Line 38-40.

Comments 4: [you can make the introduction section related to your research study .]
Response 4: Thank you for your suggestion. We modified it in Line 81-85.

Comments 5: [you can remove unnecessary sentences in the results section as you have already mentioned results in the table. Table 1 presents the characteristics of study subjects across the Treatment (N=35) and 81 Placebo (N=15) groups. The mean age of the treatment group is 44.34 years (SD=8.93),  while the placebo group has a mean age of 43.07 years (SD=8.40), with a p-value of 0.64.  The average frequency of BQ use is 5.54 days per week (SD=2.33) in the treatment group 
and 6.13 days per week (SD=1.81) in the placebo group, with a p-value of 0.39. The mean 
amount of BQ used in the treatment group is 50.97 (SD=55.73), compared to 36.27 (SD=39.64) in the placebo group, with a p-value of 0.36. Regarding DSM 5 criteria for BQ,  tobacco, and alcohol, as well as SUSRS]
Response 5: Thank you for pointing out these valuable opinions. We modified the table 1 in Line 91-97, table 2 in Line 107-108, table 3 in Line 121-122, table 4 in Line 138-139, and table 5 in Line 153-155 accordingly.

Comments 6: [after following you can mention it:  It has been widely applied in assessing alcohol consumption, cigarette smoking, and drug use[17, 18]. Smoking is one of the confounding factors in our study since smoking can impact symptom severity of psychiatric problems (Uludag&Zhao, 2023). Uludag, K., & Zhao, M. (2023). A narrative review on the association between smoking and schizophrenia symptoms. J Clin Basic Psychosom, 1(1), 1014.]
Response 6:  Your advice is very helpful and we're appreciated about this so much. We added the reference and modified it in Line 274-276.

Comments 7: [if possible you can write more about inclusion and exclusion criteria of your study.]
Response 7:  Thank you for your good advices and we modified them again. The inclusions criteria are in Line 238-245 and exclusion criteria in Line 246-259.

Comments 8: [you can mention low sample size as one of the limitations of your study.]
Response 8: Thank you for your suggestion and we modified it in Line 224-227.

Comments 9: [you can add figures related to your study. ]
Response 9:  Thank you so much for the comprehensive suggestions. Due to limitation of the numbers of tables and figures, we performed the figures related to our study in supplementary figure 1-5. The five figures were added as the supplementary materials. And we believed the supplementary materials also convinced our novel finding that different MAOA genotypes exhibit varying responses to antidepressant treatment in BUD patients and rendering MAOA genotypes as potential biomarkers for betel-quid addiction.

Comments 10: [you should make explanations about following as you claimed they are not cancer patients. Also, if possible explain how many cancer patients were excluded: The study recruited participants from the cancer centers of the Department of Dentistry and the Department of General Physicians at China Medical University Hospital inTaichung, Taiwan, between January 2016 and April 2019.]
Response 10:  Thank you again for the comprehensive suggestion.  In the manuscript, the exclusion criteria include any forms of cancer or cancer-related diseases. We did not recruit any subjects with a history of cancer. We modified it and listed some examples of cancers excluded out in Line 252-254.